# Learning from Partially-Observed Multimodal Data with Variational Autoencoders

## Abstract

Learning from only partially-observed data for imputation has been an active research area. Despite promising progress on unimodal data imputation (e.g., image in-painting), models designed for multimodal data imputation are far from satisfactory. In this paper, we propose variational selective autoencoders (VSAE) for this task. Different from previous works, our proposed VSAE learns only from partially-observed data. VSAE is capable of learning the joint distribution of observed and unobserved modalities as well as the imputation mask, resulting in a unified model for various down-stream tasks including data generation and imputation. Evaluation on both synthetic high-dimensional and challenging low-dimensional multi-modality datasets shows significant improvement over the state-of-the-art data imputation models.

## 1 Introduction

Learning from data is an integral part of machine learning and artificial intelligence. Modern deep learning techniques rely heavily on extracting information form large scale datasets. While such frameworks have been shown to be effective on various down-stream tasks such as classification, regression, representation learning, and prediction, it is typically crucial to have access to clean and complete training data. Complete data in this case can be either labeled data (for classification), or time-series data with no missing values (for regression), or simply image with no missing pixels (for generation). As such, if a model can only access partially-observed data, the performance will likely be much worse than those trained with fully-observed data, if not completely failing. In practical scenarios, however, it is usually costly to acquire clean and complete data due to the limited human resources and time. Having a model designed to learn and extract information from partially-observed data will not only largely increase the application spectrum of deep learning based models, but also provide benefit to new down-stream tasks, for example, data imputation.

Data imputation with deep generative models has been an active research area (Yoon et al., 2018; Ivanov et al., 2019; Nazabal et al., 2018). Despite promising progress, there are still challenges in learning effective models. First, some prior works focus on learning from fully-observed data while performing imputation on partially-observed data during test phase (Suzuki et al., 2016; Ivanov et al., 2019). Second, they usually have strong assumptions on missingness mechanism (see A.1) such as data is missing completely at random (MCAR) (Yoon et al., 2018). Third, mostly unimodal imputation such as image in-painting has been explored for high-dimensional data (Ivanov et al., 2019; Mattei & Frellsen, 2019). Unimodal data refers to data with only one modality such as image, video, or text. Modeling any combination of data modalities is not well-established yet, which apparently limits the potential of such models, since raw data in real-life is usually acquired in a multimodal manner (Ngiam et al., 2011) with more than one source of data gathered to represent a practical scenario. In practice, one or more of the modalities maybe be missing, leading to a challenging multimodal data imputation task.

In this work, we propose Variational Selective Autoencoder (VSAE) for multimodal data generation and imputation. Our proposed VSAE tries to address the challenges above by learning from partially-observed training data. By constructing an encoder for each modality independently, the latent representation selectively takes only the observed modalities as input, while a set of decoders maps the latent codes to not only *full data* (including both observed and unobserved modalities), but also a *mask* representing the missingness scheme. Thus, it can model the joint distribution of the data

and the mask together and avoid limiting assumptions such as MCAR, and is optimized efficiently with a single variational objective. In our experimental validation, we evaluate our proposed VSAE on both synthetic high-dimensional multimodal data and challenging low-dimensional tabular data, and show that VSAE can outperform state-of-the-art baseline models for data imputation task. The contributions are summarized as follows:

(1) A novel framework VSAE to learn from partially-observed multimodal data.

(2) The proposed VSAE is capable of learning the joint distribution of observed and unobserved modalities as well as the imputation mask, resulting in a unified model for various down-stream tasks including data generation and imputation with relaxed assumptions on missingness mechanism.

(3) Evaluation on both synthetic high-dimensional and challenging low-dimensional multimodal datasets shows improvement over the state-of-the-art data imputation models.

## 2 RELATED WORK

Our work is related to literature on *data imputation* and *multi-modal representation learning*. In this section, we briefly review recent models proposed in these two domains.

**Data Imputation.** Classical imputation methods such as MICE (Buuren & Groothuis-Oudshoorn, 2010) and MissForest (Stekhoven & Bühlmann, 2011) learn discriminative models to impute missing features from observed ones. With recent advances in deep learning, several deep imputation models have been proposed based on autoencoders (Vincent et al., 2008; Gondara & Wang, 2017; Ivanov et al., 2019), generative adversarial nets (GANs) (Yoon et al., 2018; Li et al., 2019), and autoregressive models (Bachman & Precup, 2015). GAN-based imputation method GAIN proposed by Yoon et al. (2018) assumes that data is missing completely at random. Moreover, this method does not scale to high-dimensional multimodal data. Several VAE based methods (Ivanov et al., 2019; Nazabal et al., 2018; Mattei & Frellsen, 2019) have been proposed in recent years. Ivanov et al. (2019) formulated VAE with arbitrary conditioning (VAEAC) which allows generation of missing data conditioned on any combination of observed data. This algorithm needs complete data during training and cannot learn from partially-observed data only. Nazabal et al. (2018) and Mattei & Frellsen (2019) modified VAE formulation to model the likelihood of the observed data only. However, they require training of a separate generative network for each dimension thereby increasing computational requirements. In contrast, our method aims to model joint distribution of observed and unobserved data along with the missing pattern (imputation mask). This enables our model to perform both data generation and imputation even under relaxed assumptions on missingness mechanism (see Appendix A.1).

**Learning from Multimodal Data.** A class of prior works such as conditional VAE (Sohn et al., 2015) and conditional multimodal VAE (Pandey & Dukkipati, 2017) focus on learning the conditional likelihood of the modalities. However, these models requires complete data during training and cannot handle arbitrary conditioning. Alternatively, several generative models aim to model joint distribution of all modalities (Ngiam et al., 2011; Srivastava & Salakhutdinov, 2012; Sohn et al., 2014; Suzuki et al., 2016). However, multimodal VAE based methods such as joint multimodal VAE (Suzuki et al., 2016) and multimodal factorization model (MFM) (Tsai et al., 2019) require complete data during training. On the other hand, Wu & Goodman (2018) proposed another multimodal VAE (namely MVAE) can be trained with incomplete data. This model leverages a shared latent space for all modalities and obtains an approximate joint posterior for the shared space assuming each modalities to be factorized. However, if training data is complete, this model cannot learn the individual inference networks and consequently does not learn to handle missing data during test. Building over multimodal VAE approaches, our model aims to address the shortcomings above within a flexible framework. In particular, our model can learn multimodal representations from partially observed training data and perform data imputation from arbitrary subset of modalities during test. By employing a factorized multimodal representations in the latent space it resembles disentangled models which can train factors specialized for learning from different parts of data (Tsai et al., 2019).

## 3 METHOD

In this section, we introduce a novel VAE-based framework named Variational Selective Autoencoder (VSAE) to learn from partially-observed multimodal data. We first formalize our problem and then provide a detailed description of our model.

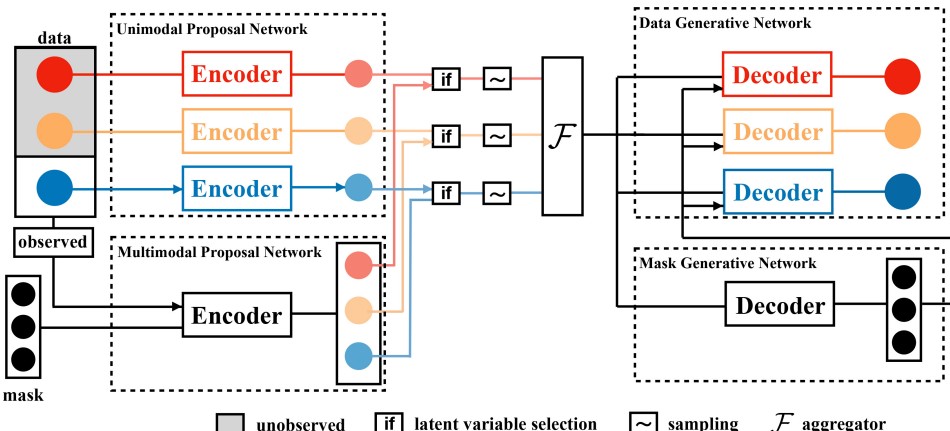

Figure 1: **Overall architecture.** The unimodal proposal network and multimodal proposal network are employed by selection. Modalities are denoted by different colors. Unobserved modalities are shaded. (i.e. blue is observed while red/yellow are unobserved.) The selected variables are indicated by the arrows. Standard normal prior is not plotted for simplicity. All components are trained simultaneously in an end-to-end manner.

### 3.1 PROBLEM STATEMENT

Let $\mathbf{x} = [\mathbf{x}_1, \mathbf{x}_2..., \mathbf{x}_M]$ be the complete data with $M$ modalities, where $\mathbf{x}_i$ denotes the feature representation for the $i$-th modality. The size of each $\mathbf{x}_i$ varies and can be very high-dimensional (*e.g.* multimedia data) or low-dimensional (*e.g.* tabular data). We define an $M$-dimensional binary mask variable $\mathbf{m} \in \{0,1\}^M$ to represent the observed and unobserved modalities: $m_i = 1$ if the $i$-th modality is observed and $0$ if unobserved. Thus we have the set of *observed modalities* $\mathbb{O} = \{i|m_i = 1\}$, and the set of *unobserved modalities* $\mathbb{U} = \{i|m_i = 0\}$. $\mathbb{O}$ and $\mathbb{U}$ are complementary subsets of all modalities. Accordingly, we denote the representation for the observed and unobserved modalities with $\mathbf{x_o} = [\mathbf{x}_i|m_i = 1]$ and $\mathbf{x_u} = [\mathbf{x}_i|m_i = 0]$, respectively. In this paper, we assume the data $\mathbf{x}$ and the mask $\mathbf{m}$ are dependent, and aim to model the joint distribution of them together.

As a result of such joint modeling, VSAE has higher capacity and can be used for both data imputation and data/mask generation. We encoder the multimodal data to a latent space factorized with respect to the modalities. To handle training and test with partially-observed data, the variational latent variable of each modality is modeled selectively to choose between a unimodal encoder if the corresponding modality is observed, or a multimodal encoder if the modality is unobserved. In addition, all the modalities and mask are reconstructed by decoding the aggregated latent codes through decoders.

### 3.2 BACKGROUND: VARIATIONAL AUTOENCODER

VAE (Kingma & Welling, 2013) is a probabilistic latent variable model to generate a random variable $\mathbf{x}$ from a latent variable $\mathbf{z}$ with a prior distribution $p(\mathbf{z})$ according to the marginalized distribution $p(\mathbf{x}) = \mathbb{E}_{\mathbf{z} \sim p(\mathbf{z})} p(\mathbf{x}|\mathbf{z}) = \int p(\mathbf{x}|\mathbf{z})p(\mathbf{z})d\mathbf{z}$. However, this is computationally intractable, so the likelihood $\log p(\mathbf{x})$ is approximated by variational lower bound (ELBO) $\mathcal{L}_{\boldsymbol{\theta},\boldsymbol{\phi}}(\mathbf{x})$:

$$\log p(\mathbf{x}) \geq \mathcal{L}_{\boldsymbol{\theta},\boldsymbol{\phi}}(\mathbf{x}) = \mathbb{E}_{\mathbf{z} \sim q_{\boldsymbol{\phi}}(\mathbf{z}|\mathbf{x})}[\log p_{\boldsymbol{\theta}}(\mathbf{x}|\mathbf{z})] - D_{\mathrm{KL}}[q_{\boldsymbol{\phi}}(\mathbf{z}|\mathbf{x})||p(\mathbf{z})]. \tag{1}$$

In this equation, $q_{\boldsymbol{\phi}}(\mathbf{z}|\mathbf{x})$ is a proposal distribution to approximate intractable true posterior $p(\mathbf{z}|\mathbf{x})$ and parameterized by an inference network (a.k.a encoder). $p_{\boldsymbol{\theta}}(\mathbf{x}|\mathbf{z})$ is the conditional likelihood parameterized by another generative network (a.k.a decoder). $D_{\mathrm{KL}}$ is the Kullback-Leibler (KL) divergence between the prior and the proposal distribution and functions as a regularizer term, $D_{\mathrm{KL}}[q_{\boldsymbol{\phi}}(\mathbf{z}|\mathbf{x})||p(\mathbf{z})] = \mathbb{E}_{\mathbf{z} \sim q_{\boldsymbol{\phi}}(\mathbf{z}|\mathbf{x})}[\log q_{\boldsymbol{\phi}}(\mathbf{z}|\mathbf{x}) - \log p(\mathbf{z})]$. To train this model $\mathcal{L}_{\boldsymbol{\theta},\boldsymbol{\phi}}(\mathbf{x})$ is optimized over all training data with respect to the parameters $\boldsymbol{\theta}$ and $\boldsymbol{\phi}$. For more details see Appendix A.2.

### 3.3 Proposed Model: Variational Selective Autoencoder

Our goal is to model the joint distribution $p(\mathbf{x}, \mathbf{m}) = \int p(\mathbf{x}, \mathbf{m}|\mathbf{z})p(\mathbf{z})d\mathbf{z}$ where $\mathbf{x} = [\mathbf{x_o}, \mathbf{x_u}]$. Following VAE formulation, we construct a proposal distribution $q(\mathbf{z}|\mathbf{x}, \mathbf{m})$ to approximate the intractable true posterior. See the architecture in Figure 1, we denote the parameters of encoder by $\{\phi, \psi\}$, and decoders of data and mask by $\theta$ and $\epsilon$ respectively. A lower bound of $\log p(\mathbf{x}, \mathbf{m})$ can be derived as:

$$
\begin{aligned}
\mathcal{L}_{\phi,\psi,\theta,\epsilon}(\mathbf{x}, \mathbf{m}) &= \mathbb{E}_{\mathbf{z}\sim q_{\phi,\psi}(\mathbf{z}|\mathbf{x},\mathbf{m})}[\log p_{\theta,\epsilon}(\mathbf{x}, \mathbf{m}|\mathbf{z})] - D_{\mathrm{KL}}[q_{\phi,\psi}(\mathbf{z}|\mathbf{x}, \mathbf{m})||p(\mathbf{z})] \\
&= \mathbb{E}_{\mathbf{z}\sim q_{\phi,\psi}(\mathbf{z}|\mathbf{x},\mathbf{m})}[\log p_{\theta}(\mathbf{x}|\mathbf{m}, \mathbf{z}) + \log p_{\epsilon}(\mathbf{m}|\mathbf{z}) - \log q_{\phi,\psi}(\mathbf{z}|\mathbf{x}, \mathbf{m}) + \log p(\mathbf{z})].
\end{aligned}
$$
(2)

We assume the variational latent variables can be factorized with respect to the modalities $\mathbf{z} = [\mathbf{z}_1, \mathbf{z}_2, ..., \mathbf{z}_M]$, which is a standard assumption for multimodal data (Tsai et al., 2019):

$$
p(\mathbf{z}) = \prod_{i=1}^{M} p(\mathbf{z}_i), \qquad q(\mathbf{z}|\mathbf{x}, \mathbf{m}) = \prod_{i=1}^{M} q(\mathbf{z}_i|\mathbf{x}, \mathbf{m}).
$$
(3)

Given this, we define the proposal distribution parameterized by $\phi$ and $\psi$ for each modality as

$$
q_{\phi,\psi}(\mathbf{z}_i|\mathbf{x}, \mathbf{m}) = \begin{cases} q_{\phi}(\mathbf{z}_i|\mathbf{x}_i) & \text{if } m_i = 1 \\ q_{\psi}(\mathbf{z}_i|\mathbf{x_o}, \mathbf{m}) & \text{if } m_i = 0 \end{cases}
$$
(4)

This is based on the intuitive assumption that the latent space of each modality is independent of other modalities given its data is observed. But, if the data is missing for some modality, its latent space is constructed from the other observed modalities. We call this *selective proposal distribution*.

In the decoder, the probability distribution also factorizes over the modalities assuming that the reconstructions are conditionally independent given the complete set of latent variables of all modalities:

$$
\log p_{\theta}(\mathbf{x}|\mathbf{m}, \mathbf{z}) = \log p_{\theta}(\mathbf{x_o}, \mathbf{x_u}|\mathbf{m}, \mathbf{z}) = \sum_{i\in\mathbb{O}} \log p_{\theta}(\mathbf{x}_i|\mathbf{m}, \mathbf{z}) + \sum_{j\in\mathbb{U}} \log p_{\theta}(\mathbf{x}_j|\mathbf{m}, \mathbf{z})
$$
(5)

To summarize, the ELBO in Equation 2 can be rewritten as

$$
\begin{aligned}
\mathcal{L}_{\phi,\psi,\theta,\epsilon}(\mathbf{x_o}, \mathbf{x_u}, \mathbf{m}) =& \mathbb{E}_{\mathbf{z}}\left[\sum_{i\in\mathbb{O}} \log p_{\theta}(\mathbf{x}_i|\mathbf{m}, \mathbf{z}) + \sum_{j\in\mathbb{U}} \log p_{\theta}(\mathbf{x}_j|\mathbf{m}, \mathbf{z})\right] + \mathbb{E}_{\mathbf{z}}[\log p_{\epsilon}(\mathbf{m}|\mathbf{z})] \\
& - \sum_{i=1}^{M} \mathbb{E}_{\mathbf{z}_i}[\log q_{\phi,\psi}(\mathbf{z}_i|\mathbf{x}, \mathbf{m}) - \log p(\mathbf{z}_i)],
\end{aligned}
$$
(6)

where $\mathbf{z}_i \sim q_{\phi,\psi}(\mathbf{z}_i|\mathbf{x}, \mathbf{m})$ according to the selective proposal distribution given in Equation 4.

For training the model, the ELBO should be maximized over training data. However under partially-observed setting, $\mathbf{x_u}$ is missing and unavailable even during training. Thus, we define the objective function for training by taking expectation over $\mathbf{x_u}$

$$
\mathcal{L}'_{\phi,\psi,\theta,\epsilon}(\mathbf{x_o}, \mathbf{m}) = \mathbb{E}_{\mathbf{x_u}}[\mathcal{L}_{\phi,\psi,\theta,\epsilon}(\mathbf{x_o}, \mathbf{x_u}, \mathbf{m})]
$$
(7)

Only one term in Equation 6 is dependent on $\mathbf{x_u}$, so the final objective function is obtained as

$$
\begin{aligned}
\mathcal{L}'_{\phi,\psi,\theta,\epsilon}(\mathbf{x_o}, \mathbf{m}) =& \mathbb{E}_{\mathbf{z}}\left[\sum_{i\in\mathbb{O}} \log p_{\theta}(\mathbf{x}_i|\mathbf{m}, \mathbf{z}) + \sum_{j\in\mathbb{U}} \mathbb{E}_{\mathbf{x}_j}[\log p_{\theta}(\mathbf{x}_j|\mathbf{m}, \mathbf{z})]\right] + \mathbb{E}_{\mathbf{z}}[\log p_{\epsilon}(\mathbf{m}|\mathbf{z})] \\
& - \sum_{i=1}^{M} \mathbb{E}_{\mathbf{z}_i}[\log q_{\phi,\psi}(\mathbf{z}_i|\mathbf{x}, \mathbf{m}) - \log p(\mathbf{z}_i)], \text{where } \mathbf{z}_i \sim q_{\phi,\psi}(\mathbf{z}_i|\mathbf{x}, \mathbf{m})
\end{aligned}
$$
(8)

In the proposed algorithm, we approximate $\mathbb{E}_{\mathbf{x}_j}[\log p_{\theta}(\mathbf{x}_j|\mathbf{m}, \mathbf{z})], j \in \mathbb{U}$ using reconstructed unobserved data sampling from the prior network. Our experiments show that even a single sample is sufficient to learn the model effectively. In fact, the prior network can be used as a self supervision mechanism to find the most likely samples which dominate the other samples when taking the expectation. In Equation 8, $p_{\theta}(\mathbf{x}_i|\mathbf{m}, \mathbf{z})$ is the decoding term of corresponding modality $\mathbf{x}_i$ and the type of distribution depends on the data. The mask decoding term $p_{\theta}(\mathbf{m}|\mathbf{z})$ is factorized Bernoulli distribution modeling the binary mask variable. The prior is standard normal distribution $p(\mathbf{z}) = \prod_{i=1}^{M} p(\mathbf{z}_i) = \prod_{i=1}^{M} \mathcal{N}(\mathbf{z}_i; \mathbf{0}, \mathbf{I})$ which is fully-factorized.

## 3.4 NETWORK MODULES

We construct each module of our model using neural networks and optimize the parameters via backpropagation techniques. Following the terms in standard VAE, VSAE is composed of encoders and decoders. The architecture is shown in Figure 1. The whole architecture can be viewed as an integration of two auto-encoding structures: the top-branch data-wise encoders/decoders and the bottom-branch mask-wise encoders/decoder. The selective proposal distribution chooses between the unimodal and multimodal encoders, depending on whether the data is observed or not. The outputs of all encoders are sampled and aggregated to provide input to all the decoders. In the rest of this section we explain different modules. See Appendix B for further implementation details.

**Selective Factorized Encoders**   Standard proposal distribution of VAEs depends on the whole data and can not handle incomplete input. To overcome this, we introduce our selective proposal distribution, which is factorized w.r.t the modalities. As defined in Equation 4, the *unimodal proposal distribution* $q_{\phi}(\mathbf{z}_i|\mathbf{x}_i)$ is inferred only from each individual observed modality (modeled by a set of separate encoders parameterized by $\phi$). If the modality is unobserved, the *multimodal proposal distribution* $q_{\psi}(\mathbf{z}_i|\mathbf{x_o}, \mathbf{m})$ (a single encoder parameterized by $\psi$) is used to infer corresponding latent variables from other observed modalities and mask. Hence, the learned model can impute the missing information by combining unimodal proposal distribution of observed modalities and multimodal proposal distribution of the unobserved modalities. The condition on the mask could make the model aware of the missing pattern and help attend to observed modalities. We model all the proposal distributions as normal distributions by setting the outputs of all encoders as mean and covariance of a normal distribution. The reparameterization in standard VAE is used for end-to-end training.

**Decoding through Latent Variable Aggregator** $\mathcal{F}$   Selected and sampled from proper proposal distributions for all modalities, the variational latent codes can be fed to the downstream decoders even when the observation is incomplete. To do this, the information from different modalities are combined by aggregating their stochastic latent codes before they are decoded using a decoder: $p_{\boldsymbol{\epsilon}}(\mathbf{m}|\mathbf{z}) = p_{\boldsymbol{\epsilon}}(\mathbf{m}|\mathcal{F}(\mathbf{z})), p_{\boldsymbol{\theta}}(\mathbf{x}_i|\mathbf{z}, \mathbf{m}) = p_{\boldsymbol{\theta}}(\mathbf{x}_i|\mathcal{F}(\mathbf{z}), \mathbf{m})$. Here, we choose the aggregator $\mathcal{F}(\cdot) = \text{concat}(\cdot)$, i.e., concatenating the latent codes. One may also use other aggregation functions such as max/mean pooling or matrix fusion (Veit et al., 2018) to combine latent codes from all modalities. The decoders take the shared aggregated latent codes as input to generate data and mask.

**Mask Vector Encoding and Decoding**   The mask variable $\mathbf{m}$ is encoded into the latent space through the multimodal proposal network. The latent space is shared by the mask and data decoders. The mask decoder $\boldsymbol{\epsilon}$ is parameterized using an MLP in our implementation. We assume each dimension of the mask variable is an independent Bernoulli distribution.

**Training**   With reparameterization trick (Kingma & Welling, 2013), we can jointly optimize the objective derived in Equation 8 with respect to these parameters defined above on training set:

$$\max_{\phi, \theta, \psi, \epsilon} \mathbb{E}_{\mathbf{x_o}, \mathbf{m}}[\mathcal{L}'_{\phi, \theta, \psi, \epsilon}(\mathbf{x_o}, \mathbf{m})] \tag{9}$$

Since Equation 9 only requires the mask and observed data during training, this modified ELBO $\mathcal{L}'_{\phi, \theta, \psi, \epsilon}(\mathbf{x_o}, \mathbf{m})$ can be optimized without the presence of unobserved modalities. The KL-divergence term is calculated analytically for each factorized term. The conditional log-likelihood term is computed by negating reconstruction loss function. (See Section 4 and Appendix B.2.)

**Inference**   The learned model can be used for both data imputation and generation. For imputation, the observed modalities $\mathbf{x_o}$ and mask $\mathbf{m}$ are fed through the encoders to infer the selective proposal distributions. Then the sampled latent codes are decoded to estimate the unobserved modalities $\mathbf{x_u}$. All the modules in Figure 1 are used for imputation. For generation, since no data is available at all, the latent codes are sampled from the prior and go through the decoders to generate the data and the mask. In this way, only modules after the aggregator are used without any inference modules.

|  | Categorical(PFC) | | Numerical(NRMSE) | |
|---|---|---|---|---|
|  | Phishing | Mushroom | Yeast | Glass |
| AE | $0.348 \pm 0.002$ | $0.556 \pm 0.009$ | $0.737 \pm 0.036$ | $1.651 \pm 0.049$ |
| VAE | $0.293 \pm 0.003$ | $0.470 \pm 0.017$ | $0.468 \pm 0.003$ | $1.409 \pm 0.011$ |
| CVAE w/ mask | $0.241 \pm 0.003$ | $0.445 \pm 0.004$ | $0.470 \pm 0.001$ | $1.498 \pm 0.001$ |
| MVAE | $0.308 \pm 0.015$ | $0.586 \pm 0.019$ | $0.475 \pm 0.014$ | $1.572 \pm 0.035$ |
| VSAE (ours) | $\mathbf{0.237 \pm 0.001}$ | $\mathbf{0.396 \pm 0.008}$ | $\mathbf{0.455 \pm 0.003}$ | $\mathbf{1.312 \pm 0.021}$ |
| CVAE w/ data | $0.301 \pm 0.005$ | $0.485 \pm 0.034$ | $0.449 \pm 0.001$ | $1.380 \pm 0.045$ |
| VAEAC | $0.240 \pm 0.006$ | $0.403 \pm 0.006$ | $0.447 \pm 0.0016$ | $1.432 \pm 0.027$ |

Table 1: **Feature Imputation on UCI datasets.** Missing ratio is 0.5. Categorical and numerical datasets are respectively evaluated by PFC and NRMSE. Last two rows are trained with fully-observed data, potentially serving as an upper bound for imputation models. We show mean and standard deviation over 3 independent runs. For both lower is better.

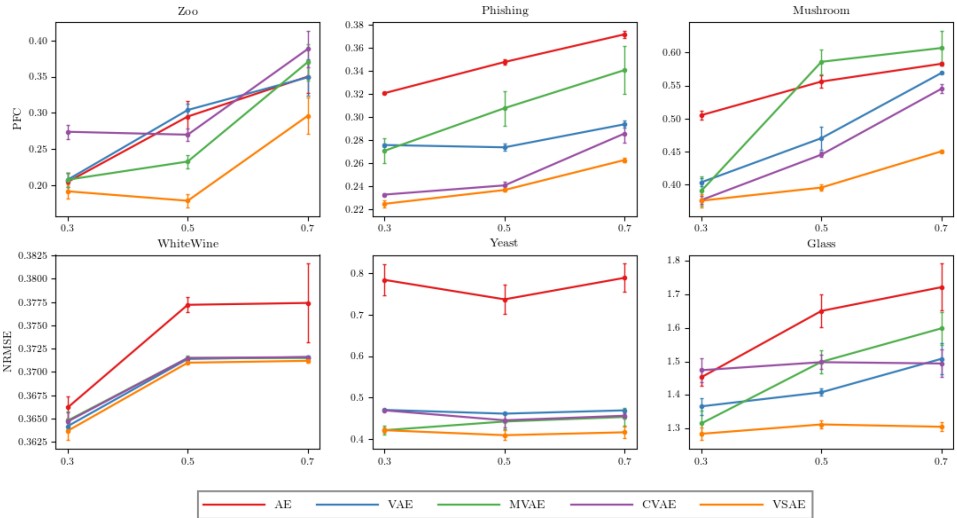

Figure 2: **Feature Imputations on UCI datasets.** Missing ratios (x-axis) are 0.3, 0.5, 0.7. Categorical (top row) and numerical (bottom row) datasets are evaluated by PFC and NRMSE respectively (lower is better for both). We show mean and standard deviation over 3 independent runs.

# 4 EXPERIMENT

To demonstrate the effectiveness of our model, we evaluate our model on *low-dimensional tabular data* imputation and *high-dimensional multi-modal data* imputation tasks, with extensive comparisons with state-of-the-art deep latent variable models.

**Baselines.** Prior work on deep latent variable models for data imputation can be categorized into two main classes: (1) models having access to fully-observed data during training, and (2) models only having access to partially observed data during training. In class (1), we report the results of VAEAC (Ivanov et al., 2019) and conditional VAE (Sohn et al., 2015); while in class (2), we report results of deterministic Autoencoder (AE), VAE (Kingma & Welling, 2013), conditional VAE (Sohn et al., 2015) (conditioned on mask) and MVAE (Wu & Goodman, 2018). Our model VSAE falls in this category since it learns the joint distribution of $p(\mathbf{x_o}, \mathbf{x_u}, \mathbf{m})$ given only observed information. Note that class (1) models can empirically represent the upper bound representative capability of imputation models, as they have access to fully-observed data during training. To establish fair comparison, all models in the experiments are implemented with the same backbone structure. Additional information on experimental details can be found in Appendix. B.

## 4.1 DATA IMPUTATION

**Low-dimensional Tabular Data Imputation.** We choose UCI repository datasets to demonstrate the effectiveness of our model on tabular data. It contains different tabular datasets with either

| | MNIST+MNIST(MSE) | | | MNIST+SVHN(MSE) | | |
|---|---|---|---|---|---|---|
| | MNIST/784 | MNIST/784 | combined | MNIST/784 | SVHN/3072 | combined |
| AE | $0.1077 \pm \Delta$ | $0.1070 \pm \Delta$ | $0.2147 \pm \Delta$ | $0.0867 \pm \Delta$ | $0.1475 \pm \Delta$ | $0.2342 \pm \Delta$ |
| VAE | $0.0734 \pm \Delta$ | $0.0682 \pm \Delta$ | $0.1396 \pm \Delta$ | $0.0714 \pm \Delta$ | $0.0559 \pm 0.003$ | $0.1273 \pm \Delta$ |
| CVAE w/ mask | $0.0733 \pm \Delta$ | $0.0679 \pm \Delta$ | $0.1412 \pm \Delta$ | $0.0692 \pm \Delta$ | $0.0558 \pm \Delta$ | $0.1251 \pm \Delta$ |
| MVAE | $0.0760 \pm \Delta$ | $0.0802 \pm \Delta$ | $0.1562 \pm \Delta$ | $0.0707 \pm \Delta$ | $0.602 \pm \Delta$ | $0.1309 \pm \Delta$ |
| VSAE (ours) | $\mathbf{0.0712 \pm \Delta}$ | $\mathbf{0.0663 \pm \Delta}$ | $\mathbf{0.1376 \pm \Delta}$ | $\mathbf{0.0682 \pm \Delta}$ | $\mathbf{0.0516 \pm \Delta}$ | $\mathbf{0.1198 \pm \Delta}$ |
| CVAE w/ data | $0.0694 \pm \Delta$ | $0.0646 \pm \Delta$ | $0.1340 \pm \Delta$ | $0.0716 \pm \Delta$ | $0.0550 \pm \Delta$ | $0.1266 \pm \Delta$ |
| VAEAC | $0.0693 \pm \Delta$ | $0.0645 \pm \Delta$ | $0.1338 \pm \Delta$ | $0.0682 \pm \Delta$ | $0.0523 \pm \Delta$ | $0.1206 \pm \Delta$ |

Table 2: **Imputation on Bimodal datasets.**. Missing ratio is 0.5. Last two rows are trained with fully-observed data. We show mean and standard deviation over 3 independent runs (lower is better). $\Delta < 0.001$.

numerical or categorical variables. In our experiments, we randomly sample from independent Bernoulli distributions with pre-defined missing ratio to simulate the masking mechanism. Min-max normalization is then applied to pre-process the numerical data and replace the unobserved dimensions by standard normal noise. We split training/test set by $80\%/20\%$ and $20\%$ of training set as validation set to choose the best model. Mean Square Error, Cross Entropy and Binary Cross Entropy are used as reconstruction loss for numerical, categorical and mask variables, respectively. We report the standard measures: **NRMSE** (i.e. RMSE normalized by the standard deviation of the feature and averaged over all features) for numerical datasets and **PFC** (i.e. proportion of falsely classified attributes of each feature and averaged over all features) for categorical datasets.

**Results and Analysis.** Table 1 shows that VSAE outperforms other methods on both numerical and categorical data. The first five rows are trained in partially-observed setting, while the last two trained with fully-observed data. We observe that models trained with partially-observed data can outperform those models trained with fully-observed data on some datasets. We argue this is due to two potential reasons: (1) the mask provides a natural way of dropout on the data space, thereby, helping the model to generalize; (2) if the data is noisy or has outliers (which is common in low-dimensional data), learning from partially-observed data can improve performance by ignoring these data. However, although our model does not product state-of-the-art results in fully-observed data imputation settings, these models potentially can serve as upper bound if the data is clean.

Figure 2 illustrates that our model generally has lower error with lower variance for all missing ratios. With higher missing ratio (more data is unobserved), our model achieves more stable imputation performance on most of the datasets. On the contrary, there is a performance drop along with higher variance in the case of baselines. We believe this is because of the proposal distribution selection in VSAE. As the missing ratio increases, the input to unimodal encoders stays same while other encoders have to learn to focus on the useful information in data.

**High-dimensional Multimodal Data.** We synthesize two bimodal datasets using MNIST and SVHN datasets. MNIST contains 28-by-28 gray images (0-9 digits); SVHN contains 32-by-32 RGB images (0-9 digits). We synthesize our datasets by pairing two different digits in MNIST (named **MNIST+MNIST**) and one digit in MNIST with a same digit in SVHN (named **MNIST+SVHN**). See Appendix C for more experimental results on multimodal FashionMNIST, MNIST and CMU-MOSI.

**Results and Analysis.** VSAE has better performance for imputation task on all modalities with lower variance (refer to Table 2). Figure 3 presents the qualitative results of imputations on MNIST+MNIST. With masks sampled with different missing ratios, the combined errors on MNIST+MNIST (i.e. sum of MSE in each modality averaged over its dimensions) of our model are $0.1371 \pm 0.0001$, $0.1376 \pm 0.0002$ and $0.1379 \pm 0.0001$ under missing ratio of 0.3, 0.5 and 0.7 (Additional results are in Appendix C.2). This indicates that VSAE is robust under different missing ratios, whereas other baselines are sensitive to the missing ratio. We believe this is because of the underlying mechanism of proper proposal distribution selection. The separate structure of unimodal/multimodal encoders helps VSAE to attend to the observed data. It limits the input of unimodal encoders to observed single modality. Thus it is more robust to the missingness. In contrast, baseline methods have only one single proposal distribution inferred from the whole input. VSAE can easily ignore unobserved noisy modalities and attends on observed useful modalities, while baselines rely on neural networks to learn useful information from the whole data (which is dominated by missing information in

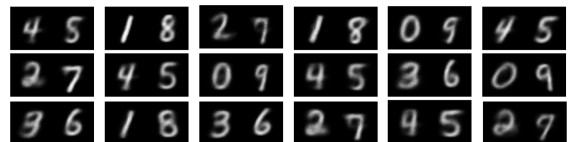

| Figure 3: **Imputation on MNIST+MNIST.** Top row visualizes observed modality, middle row unobserved modality, and bottom row shows the imputation of unobserved modality from VSAE. | Figure 4: **Generation on MNIST+MNIST.** Generated Samples w/o conditional information. As shown, the correspondence between modalities (predefined pairs) are preserved while stochastic multimodal generation. |

case of high missing ratio). For partially-observed training setting, unobserved data is not available even during training. However, the unobserved modality in one data sample could be the observed modality in another data sample. Thus, the multimodal encoders are able to construct the mapping from observable to unobservable information over the whole training set. Multimodal encoders also include the mask vector as input. This allows the multimodal encoders to be aware of the shape of the missingness and forces it to focus on the useful information in the observed modalities.

### 4.2 IMPUTATION ON NON-MCAR MASKING MECHANISMS

Sampling mask on predefined missing ratio is MCAR. VSAE can model mask distribution w/o constraints on the masking mechanisms. We also evaluate our model on MAR and NMAR. Mattei & Frellsen (2019) synthesize MAR in a defined rule and we follow them to synthesize both MAR and NMAR (refer to Appendix C.4 for details). Our model can outperform state-of-the-art non-MCAR model MIWAE (Mattei & Frellsen, 2019).

|      | MIWAE | VSAE |
| --- | --- | --- |
| MCAR | $0.467 \pm \Delta$ | $\mathbf{0.455 \pm \Delta}$ |
| MAR | $0.493 \pm 0.03$ | $\mathbf{0.472 \pm 0.02}$ |
| NMAR | $0.513 \pm 0.04$ | $\mathbf{0.456 \pm \Delta}$ |

Table 3: **Imputation.** NRMSE on Yeast. Lower is better. $\Delta < 0.01$.

### 4.3 DATA AND MASK GENERATION

Unlike conventional methods modeling $p(\mathbf{x_u}|\mathbf{x_o})$, our method is to model the joint probability $p(\mathbf{x_o}, \mathbf{x_u}, \mathbf{m})$. Thus our model can impute missing features and also generate data and mask from scratch. Figure 4 shows the model learns the correlation between different modalities to pair the digits as predefined in the dataset without giving any labels in partially-observed setting.

Our proposed VSAE can also learn to generate mask. The objective ELBO has a mask conditional log-likelihood term. This allows the latent space to have information from mask variables and be able to reconstruct (or generate if sample the prior) the mask vector. In UCI repository experiments, the mask variable follows Bernoulli distribution. After training, we sample from the prior to generate the mask. We calculate the proportion of the unobserved dimensions in generated mask vectors (averaged over 100 samples of the output). Averaged on all datasets, this proportion is $0.3123 \pm 0.026$, $0.4964 \pm 0.005$, $0.6927 \pm 0.013$ for missing ratio of 0.3, 0.5, 0.7. It indicates that our model can learn the mask distribution. We also observe that conditions on the reconstructed mask vector in the data decoders improve the performance. We believe this is because the mask vector can inform the data decoder about the missingness in the data space since the latent space is shared by both all modalities thereby allowing it to generate data from the selective proposal distribution.

## 5 CONCLUSION

In this paper, we propose a VAE framework to learn from partially-observed data. Learning from partially-observed data is important but previous deep latent variable models cannot work well on this problem. The proposed model differentiates the observed and unobserved information by selecting a proper proposal distribution. The experimental results show the model can consistently outperform other baselines on low-dimensional tabular data and high-dimensional multimodal data. The model can also generate data with mask directly from prior without any conditions.

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

# A  BACKGROUND

## A.1  IMPUTATION PROCESS AND MISSINGNESS MECHANISMS

Following (Little & Rubin, 1986), the imputation process is to learn a generative distribution for unobserved missing data. To be consistent with notations in Section **??**, let $\mathbf{x} = [\mathbf{x}_1, \mathbf{x}_2, ..., \mathbf{x}_M]$ be the complete data of all modalities where $\mathbf{x}_i$ denote the feature representation for the $i$-th modality. We also define $\mathbf{m} \in \{0,1\}^M$ as the binary mask vector, where $m_i = 1$ indicates if the $i$-th modality is observed, and $m_i = 0$ indicates if it is unobsrved:

$$\mathbf{x} \sim p_{\text{data}}(\mathbf{x}),$$
$$\mathbf{m} \sim p(\mathbf{m}|\mathbf{x}). \tag{10}$$

Given this, the observed data $\mathbf{x_o}$ and unobserved data $\mathbf{x_u}$ are represented accordingly:

$$\mathbf{x_o} = [\mathbf{x}_i|m_i = 1],$$
$$\mathbf{x_u} = [\mathbf{x}_i|m_i = 0]. \tag{11}$$

In the standard maximum likelihood setting, the unknown parameters are estimated by maximizing the following marginal likelihood, integrating over the unknown missing data values:

$$p(\mathbf{x_o}, \mathbf{m}) = \int p(\mathbf{x_o}, \mathbf{x_u}, \mathbf{m})d\mathbf{x_u} = \int p(\mathbf{x_o}, \mathbf{x_u})p(\mathbf{m}|\mathbf{x_o}, \mathbf{x_u})d\mathbf{x_u} \tag{12}$$

Little & Rubin (1986) characterizes the missingness mechanism $p(\mathbf{m}|\mathbf{x_o}, \mathbf{x_u})$ in terms of independence relations between the complete data $\mathbf{x} = [\mathbf{x_o}, \mathbf{x_u}]$ and the mask $\mathbf{m}$:

- Missing completely at random (MCAR): $p(\mathbf{m}|\mathbf{x_o}, \mathbf{x_u}) = p(\mathbf{m})$,
- Missing at random (MAR): $p(\mathbf{m}|\mathbf{x_o}, \mathbf{x_u}) = p(\mathbf{m}|\mathbf{x_o})$,
- Not missing at random (NMAR): $p(\mathbf{m}|\mathbf{x_o}, \mathbf{x_u}) = p(\mathbf{m}|\mathbf{x_u})$ or $p(\mathbf{m}|\mathbf{x_o}, \mathbf{x_u})$.

Most previous data imputation methods works under MCAR or MAR assumptions since $p(\mathbf{x_o}, \mathbf{m})$ can be factorized into $p(\mathbf{x_o})p(\mathbf{m}|\mathbf{x_o})$ or $p(\mathbf{x_o})p(\mathbf{m})$. With such decoupling, we do not need missing information to marginalize the likelihood, and it provides a simple but approximate framework to learn from partially-observed data.

## A.2  VARIATIONAL AUTOENCODER

Variational Autoencoder (VAE) (Kingma & Welling, 2013) is a probabilistic generative model, where data is constructed from a latent variable $\mathbf{z}$ with a prior distribution $p(\mathbf{z})$. It is composed of an inference network and a generation network to encode and decode data. To model the likelihood of data, the true intractable posterior $p(\mathbf{z}|\mathbf{x})$ is approximated by a proposal distribution $q_\phi(\mathbf{z}|\mathbf{x})$, and the whole model is trained until ideally the decoded reconstructions from the latent codes sampled from the approximate posterior match the training data. In the generation module, $p_\theta(\tilde{\mathbf{x}}|\mathbf{z})$, a decoder realized by a deep neural network parameterized by $\theta$, maps a latent variable $\mathbf{z}$ to the reconstruction $\tilde{\mathbf{x}}$ of observation $\mathbf{x}$. In the inference module, an encoder parameterized by $\phi$ produces the sufficient statistics of the approximation posterior $q_\phi(\mathbf{z}|\mathbf{x})$ (a known density family where sampling can be readily done). In vanilla VAE setting, by simplifying approximate posterior as a parameterized diagonal normal distribution and prior as a standard diagonal normal distribution $\mathcal{N}(\mathbf{0}, \mathbf{I})$, the training criterion is to maximize the following evidence lower bound (ELBO) w.r.t. $\theta$ and $\phi$.

$$\log p(\mathbf{x}) \geq \mathcal{L}_{\theta,\phi}(\mathbf{x}) = \mathbb{E}_{q_\phi(\mathbf{z}|\mathbf{x})}[\log p_\theta(\mathbf{x}|\mathbf{z})] - D_{\text{KL}}[q_\phi(\mathbf{z}|\mathbf{x})||p(\mathbf{z})] \tag{13}$$

where $D_{\text{KL}}$ denotes the Kullback-Leibler (KL) divergence. Usually the prior $p(\mathbf{z})$ and the approximate $q_\phi(\mathbf{z}|\mathbf{x})$ are chosen to be in simple form, such as a Gaussian distribution with diagonal covariance, which allows for an analytic calculation of the KL divergence. While VAE approximates $p(\mathbf{x})$, conditional VAE (Sohn et al., 2015) approximates the conditional distribution $p(\mathbf{x}|\mathbf{y})$. By simply introducing a conditional input, CVAE is trained to maximize the following ELBO:

$$\log p(\mathbf{x}|\mathbf{y}) \geq \mathcal{L}_{\theta,\phi,\psi}(\mathbf{x}, \mathbf{y}) = \mathbb{E}_{q_\phi(\mathbf{z}|\mathbf{x},\mathbf{y})}[\log p_\theta(\mathbf{x}|\mathbf{z}, \mathbf{y})] - D_{\text{KL}}[q_\phi(\mathbf{z}|\mathbf{x}, \mathbf{y})||p_\psi(\mathbf{z}|\mathbf{y})] \tag{14}$$

## B  Implementation Details

### B.1  Architecture

In all models, all the layers are modeled by MLP without any skip connections or resnet modules. Basically, the unimodal encoders take single modality data vector as input to infer the unimodal proposal distribution; the multimodal encoders take the observed data vectors and mask vector as as input to infer the multimodal proposal distributions. The input vector to multimodal encoders should have same length for the neural network. Here we just concatenate all modality vectors and replace the unobserved modality vectors with some noise. In UCI repository experiment, we replace the unobserved modality vectors as standard normal noise. In Bimodal experiment, we simply replace the pixels of unobserved modality as zero. Note that all the baselines has encoders/decoders with same or larger number of parameters than our method. We implement our model using PyTorch.

**Unimodal Encoders**  In UCI repository experiment, the unimodal encoders for numerical data are modeled by 3-layer 64-dim MLPs and the unimodal encoders for categorical data are modeled by 3-layer 64-dim MLPs, all followed by Batch Normalization and Leaky ReLU nonlinear activations. In MNIST+MNIST bimodal experiment, the unimodal encoders are modeled by 3-layer 128-dim MLPs followed by Leaky ReLU nonlinear activations; In MNIST+SVHN bimodal experiment, the unimodal encoders are modeled by 3-layer 512-dim MLPs followed by Leaky ReLU nonlinear activations. We set the latent dimension as 20-dim for every modality in UCI repository experiments and 256-dim for every modality in Bimodal experiments.
UCI data unimodal encoder: Linear(1, 64)$\to$ BatchNorm1d(64)$\to$ LeakyReLU$\to$ Linear(64, 64)$\to$ LeakyReLU$\to$ Linear(64, 64)$\to$ LeakyReLU$\to$ Linear(64, 20);
MNIST+MNIST synthetic unimodal encoder: Linear(data-dimension, 128)$\to$ LeakyReLU$\to$ Linear(128,128)$\to$ LeakyReLU$\to$ Linear(128, 128)$\to$ LeakyReLU$\to$ Linear(128, 256);
MNIST+SVHN synthetic unimodal encoder: Linear(data-dimension, 512)$\to$ LeakyReLU$\to$ Linear(512,512)$\to$ LeakyReLU$\to$ Linear(512, 512)$\to$ LeakyReLU$\to$ Linear(512, 256);

**Multimodal Encoders**  In general, any model capable of multimodal fusion (Zadeh et al., 2017; Morency et al., 2011) can be used here to map the observed data $\mathbf{x_o}$ and the mask $\mathbf{m}$ to the latent variables $\mathbf{z}$. However, in this paper we simply use an architecture similar to unimodal encoders. The difference is that the input to unimodal encoders are lower dimensional vectors of an individual modalities. But, the input to the multimodal encoders is the complete data vector with unobserved modalities replaced with noise or zeros. As the input to the multimodal encoders is the same for all modalities (i.e., $q(\mathbf{z}_i|\mathbf{x_o}, \mathbf{m}) \; \forall i$), we model the multimodal encoders as one single encoder to take advantage of the parallel matrix calculation speed. Thus the multimodal encoder for every experiment has the same structure as its unmidal encoder but with full-dimensional input.

**Aggregator**  In our models, we simply use vector concatenation as the way of aggregating.

**Mask Decoder**  UCI mask decoder: Linear(20*data-dimension, 64)$\to$ BatchNorm1d(64)$\to$ LeakyReLU$\to$ Linear(64, 64)$\to$ LeakyReLU$\to$ Linear(64, 64)$\to$ LeakyReLU$\to$ Linear(64, mask-dimension)$\to$Sigmoid;
MNIST+MNIST synthetic mask decoder: Linear(512, 16)$\to$ BatchNorm1d(16)$\to$ LeakyReLU$\to$ Linear(16,16)$\to$ LeakyReLU$\to$ Linear(16, 16)$\to$ LeakyReLU$\to$ Linear(16, 2)$\to$Sigmoid;
MNIST+SVHN synthetic mask encoder: Linear(512, 16)$\to$ BatchNorm1d(16)$\to$ LeakyReLU$\to$ Linear(16,16)$\to$ LeakyReLU$\to$ Linear(16,16)$\to$ LeakyReLU$\to$ Linear(16,2)$\to$Sigmoid;

**Data Decoder**  As the output is factorized over modalities and for every decoder the input is shared as the latent codes sampled from the selective proposal distribution. We implement all the decoders of the data modalities as one single decoder for parallel speed. UCI data decoder: Linear(20*data-dimension, 128)$\to$ BatchNorm1d(128)$\to$ LeakyReLU$\to$ Linear(128)$\to$ Linear(128, 128)$\to$ Linear(128, data-dimension);
MNIST+MNIST synthetic data decoder: Linear(512, 128)$\to$ BatchNorm1d(128)$\to$ LeakyReLU$\to$ Linear(128,128)$\to$ Linear(128, 128)$\to$ Linear(128, 784)$\to$Sigmoid;
MNIST+SVHN synthetic mask encoder: Linear(512, 512)$\to$ BatchNorm1d(512)$\to$ LeakyReLU$\to$ Linear(512,512)$\to$ Linear(512,512)$\to$ Linear(512,784/3072)$\to$Sigmoid;

## B.2 TRAINING

We use Adam optimizer for all models. For UCI numerical experiment, learning rate is 1e-3 and use validation set to find a best model in 1000 epochs. For UCI categorical experiment, learning rate is 1e-2 and use validation set to find a best model in 1000 epochs. For bimodal experiments, learning rate is 1e-4 and use validation set to find a best model in 1000 epochs. All modules in our models are trained jointly.

In our model, we calculate the conditional log-likelihood of unobserved modality by generating corresponding modalities from prior. We initially train the model for some (empirically we choose 20) epochs without calculating the conditional log-likelihood of $x_u$. And then first feed the partially-observed data to the model and generate the unobserved modality $\tilde{x}_u$ without calculating any loss; then feed the same batch for another pass, calculate the conditional log-likelihood using real $x_o$ and generated $x_u$ as ground truth.

## B.3 BASELINES

In our experiments, all the baselines use the same backbone architecture as our model, and the some of the layers are widened to make the total number of parameters same as our proposed model. All baselines for each experiment are trained with same Adam optimizer with same learning rate. All the deep latent variable model baselines have same size of latent variables.

In the setting of AE/VAE, the input is the whole data representation with all the modalties without any mask information; In CVAE w/ mask, the encoder and decoder are both conditioned on the mask vector, while in CVAE w/ data, the observed modalities are fed to encoder and the decoder is conditioned on the observed modalities. VAEAC (Ivanov et al., 2019) is slightly modified to remove all the skip-connections to provide a fair comparison (we do not claim we outperform VAEAC with fully-observed training) and MVAE (Wu & Goodman, 2018) is same as the proposed model architecture.

## C  ADDITIONAL EXPERIMENTAL RESULTS

### C.1  UCI REPOSITORY DATASETS

|  | Phishing | Zoo | Mushroom |
|---|---|---|---|
| AE | $0.348 \pm 0.002$ | $0.295 \pm 0.022$ | $0.556 \pm 0.009$ |
| VAE | $0.293 \pm 0.003$ | $0.304 \pm 0.009$ | $0.470 \pm 0.017$ |
| CVAE w/ mask | $0.241 \pm 0.003$ | $0.270 \pm 0.023$ | $0.445 \pm 0.004$ |
| MVAE | $0.308 \pm 0.015$ | $0.233 \pm 0.013$ | $0.586 \pm 0.019$ |
| VSAE | $\mathbf{0.237 \pm 0.001}$ | $\mathbf{0.213 \pm 0.004}$ | $\mathbf{0.396 \pm 0.008}$ |
| CVAE w/ data | $0.301 \pm 0.005$ | $0.323 \pm 0.032$ | $0.485 \pm 0.034$ |
| VAEAC | $0.240 \pm 0.006$ | $0.168 \pm 0.006$ | $0.403 \pm 0.006$ |

Table 4: **Imputation on Categorical datasets**. Missing ratio is 0.5. Last two rows are trained with fully-observed data. Evaluated by PFC, lower is better.

|  | Yeast | White Wine | Glass |
|---|---|---|---|
| AE | $0.737 \pm 0.036$ | $0.3772 \pm 0.0008$ | $1.651 \pm 0.049$ |
| VAE | $0.468 \pm 0.003$ | $0.3714 \pm 0.0001$ | $1.409 \pm 0.011$ |
| CVAE w/ mask | $0.470 \pm 0.001$ | $0.3716 \pm 0.0001$ | $1.498 \pm 0.0013$ |
| MVAE | $0.475 \pm 0.014$ | $\mathbf{0.3722 \pm 0.0009}$ | $1.572 \pm 0.035$ |
| VSAE | $\mathbf{0.455 \pm 0.003}$ | $\mathbf{0.3711 \pm 0.0002}$ | $\mathbf{1.312 \pm 0.021}$ |
| CVAE w/ data | $0.449 \pm 0.0001$ | $0.3567 \pm 0.0016$ | $1.380 \pm 0.045$ |
| VAEAC | $0.447 \pm 0.0016$ | $0.3647 \pm 0.0039$ | $1.432 \pm 0.027$ |

Table 5: **Imputation on Numerical datasets**. Missing ratio is 0.5. Last two rows are trained with fully-observed data. Evaluated by NRMSE, lower is better.

## C.2 MNIST+MNIST BIMODAL DATASET

### C.2.1 SETUP

**MNIST+MNIST bimodal dataset.** We randomly pair two digits in MNIST as [0, 9], [1, 8], [2, 7], [3, 6], [4, 5]. The training/test/validation sets respectively contain 23257/4832/5814 samples.

### C.2.2 ADDITIONAL RESULTS

|  | 0.3 | 0.5 | 0.7 |
|---|---|---|---|
| AE | $0.2124 \pm 0.0012$ | $0.2147 \pm 0.0008$ | $0.2180 \pm 0.0008$ |
| VAE | $0.1396 \pm 0.0002$ | $0.1416 \pm 0.0001$ | $0.1435 \pm 0.0006$ |
| CVAE w/ mask | $0.1393 \pm 0.0002$ | $0.1412 \pm 0.0006$ | $0.1425 \pm 0.0012$ |
| MVAE | $0.1547 \pm 0.0012$ | $0.1562 \pm 0.0003$ | $0.1579 \pm 0.0006$ |
| VSAE | $\mathbf{0.1371 \pm 0.0001}$ | $\mathbf{0.1376 \pm 0.0002}$ | $\mathbf{0.1379 \pm 0.0001}$ |
| CVAE w/ data | $0.1336 \pm 0.0003$ | $0.1340 \pm 0.0003$ | $0.1343 \pm 0.0002$ |
| VAEAC | $0.1333 \pm 0.0004$ | $0.1338 \pm 0.0003$ | $0.1344 \pm 0.0001$ |

Table 6: **Imputation on MNIST+MNIST.** Missing ratio is 0.3, 0.5 and 0.7. Last two rows are trained with fully-observed data. Evaluated by combined errors of two modalities, lower is better.

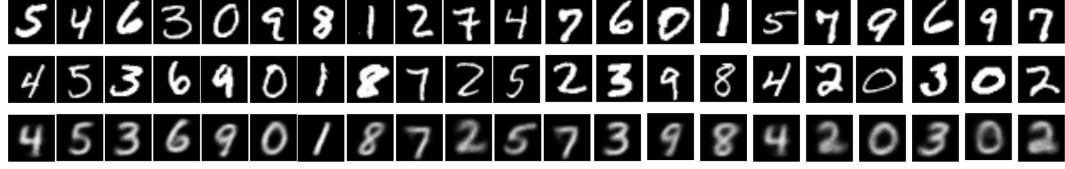

Figure 5: **Imputation on MNIST+MNIST.** Top row visualizes observed modality, middle row unobserved modality, and bottom row shows the imputation of unobserved modality from VSAE.

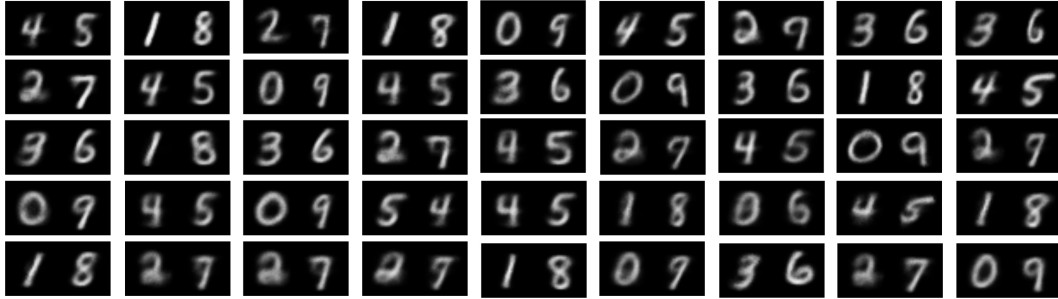

Figure 6: **Generation on MNIST+MNIST.** Generated Samples w/o conditional information. As shown, the correspondence between modalities (pre-defined pairs) are preserved while generation.

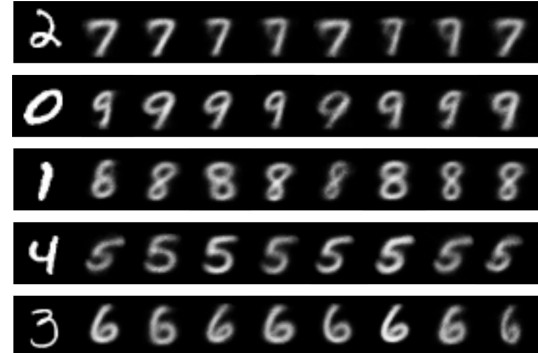

Figure 7: **Multiple independent sampling in selected latent space.** The leftmost digits are observed images in ground truth, and the right 8 digits are imputations of corresponding unobserved digits.

## C.3  MNIST+SVHN Bimodal dataset

### C.3.1  Setup

**MNIST+SVHN bimodal dataset:** We pair one digit in MNIST with the random same digit in SVHN. The training/test/validation sets respectively contain 44854/10000/11214 samples. For both datasets, we synthesize mask vectors over each modality by sampling from Bernoulli distribution. All mask are fixed after synthesis process. All original data points are only used once.

### C.3.2  Additional results

|  | MNIST-MSE/784 | SVHN-MSE/3072 | Combined Bimodal Error |
|---|---|---|---|
| AE | $0.0867 \pm 0.0001$ | $0.1475 \pm 0.0006$ | $0.2342 \pm 0.0007$ |
| VAE | $0.0714 \pm 0.0001$ | $0.0559 \pm 0.0027$ | $0.1273 \pm 0.0003$ |
| CVAE w/ mask | $0.0692 \pm 0.0001$ | $0.0558 \pm 0.0003$ | $0.1251 \pm 0.0005$ |
| MVAE | $0.0707 \pm 0.0003$ | $0.602 \pm 0.0001$ | $0.1309 \pm 0.0005$ |
| VSAE | $\mathbf{0.0682 \pm 0.0001}$ | $\mathbf{0.0516 \pm 0.0001}$ | $\mathbf{0.1198 \pm 0.0001}$ |
| CVAE w/ data | $0.0716 \pm 0.0002$ | $0.0550 \pm 0.0007$ | $0.1266 \pm 0.0005$ |
| VAEAC | $0.0682 \pm 0.0001$ | $0.0523 \pm 0.0001$ | $0.1206 \pm 0.0001$ |

Table 7: **Imputation on MNIST+SVHN.** Missing ratio is 0.5. Last two rows are trained with fully-observed data. Evaluated by combined errors of two modalities, lower is better.

|            | 0.3                    | 0.5                    | 0.7                    |
| ---------- | ---------------------- | ---------------------- | ---------------------- |
| AE         | $0.1941 \pm 0.0006$    | $0.2342 \pm 0.0007$    | $0.2678 \pm 0.0012$    |
| VAE        | $0.1264 \pm 0.0001$    | $0.1273 \pm 0.0003$    | $0.1322 \pm 0.0005$    |
| CVAE w/ mask | $0.1255 \pm 0.0002$  | $0.1251 \pm 0.0005$    | $0.1295 \pm 0.0006$    |
| MVAE       | $0.1275 \pm 0.0029$    | $0.1309 \pm 0.0005$    | $0.1313 \pm 0.0013$    |
| VSAE       | $\mathbf{0.1217 \pm 0.0002}$ | $\mathbf{0.1198 \pm 0.0001}$ | $\mathbf{0.1202 \pm 0.0002}$ |
| CVAE w/ data | $0.1288 \pm 0.0011$  | $0.1266 \pm 0.0005$    | $0.1248 \pm 0.0003$    |
| VAEAC      | $0.1218 \pm 0.0002$    | $0.1206 \pm 0.0001$    | $0.1211 \pm 0.0001$    |

Table 8: **Imputation on MNIST+SVHN.** Missing ratio is 0.3, 0.5 and 0.7. Last two rows are trained with fully-observed data. Evaluated by combined errors of two modalities, lower is better.

## C.4 IMPUTATION ON NON-MCAR MASKING MECHANISMS

VSAE can jointly model data and mask distribution without any assumption on mask distribution. See A.1 for masking mechanism definitions. Mattei & Frellsen (2019) synthesized the mask from a MAR manner. We similarly follow them to synthesize MAR/NMAR masking mechanism on UCI numerical dataset and compare to state-of-the-art non-MCAR model MIWAE (Mattei & Frellsen, 2019).

**Missing At Random (MAR).**   The mask distribution depends on the observed data. We choose first 25% modalties as default observed data and generate the mask according to the probability that

$$\pi(\mathbf{m}) = \mathrm{sigmoid}(\frac{1}{M} \sum_{k=1}^{K} \mathbf{x}_k)$$

$M$ is the number of the features and $K$ is the number of default observed features.

**Not Missing At Random (NMAR).**   The mask distribution depends on both observed and unobserved data. We generate the element-wise mask according to the probabilty that

$$\pi(m_i) = \mathrm{sigmoid}(\mathbf{x}_i)$$

$m_i$ is $i$-th element in mask vector $\mathbf{m}$ of size $M$.

|                                | MCAR                    | MAR                   | NMAR                    |
| ------------------------------ | ----------------------- | --------------------- | ----------------------- |
| MIWAE(Mattei & Frellsen, 2019) | $0.467 \pm 0.0067$      | $0.493 \pm 0.029$     | $0.513 \pm 0.035$       |
| VSAE(ours)                     | $\mathbf{0.455 \pm 0.0003}$ | $\mathbf{0.472 \pm 0.024}$ | $\mathbf{0.455 \pm 0.0001}$ |

Table 9: **Imputation on MAR/NMAR masking.** Missing ratio is based on the values of data following the defined rules above. We show mean and standard deviation over 3 independent runs (lower is better) on Yeast dataset.

## C.5 MULTIMODAL EXPERIMENT

In this section, we include additional experiments on multimodal datasets to demonstrate the general effectiveness of our model. We choose the datasets following MVAE (Wu & Goodman, 2018) and MFM Tsai et al. (2019).

|  | FashionMNIST | | MNIST | |
|---|---|---|---|---|
|  | image (MSE) | text (PFC) | image (MSE) | text (PFC) |
| AE | $86.63 \pm 1.09$ | $0.366 \pm \Delta$ | $54.90 \pm 0.01$ | $0.406 \pm \Delta$ |
| VAE | $69.38 \pm 0.10$ | $0.411 \pm \Delta$ | $53.82 \pm 0.12$ | $0.406 \pm 0.01$ |
| CVAE w/ mask | $69.53 \pm 0.65$ | $0.412 \pm \Delta$ | $53.82 \pm \Delta$ | $0.419 \pm \Delta$ |
| MVAE | $109.95 \pm 20.78$ | $0.374 \pm 0.07$ | $178.40 \pm 14.29$ | $0.448 \pm \Delta$ |
| VSAE (ours) | $\mathbf{68.49 \pm 0.19}$ | $\mathbf{0.356 \pm \Delta}$ | $\mathbf{53.42 \pm 0.05}$ | $\mathbf{0.397 \pm 0.01}$ |
| CVAE w/ data | $54.15 \pm 0.03$ | $0.259 \pm \Delta$ | $47.38 \pm \Delta$ | $0.237 \pm \Delta$ |
| VAEAC | $61.59 \pm 0.03$ | $0.283 \pm \Delta$ | $51.49 \pm 0.06$ | $0.250 \pm \Delta$ |

Table 10: **Imputation on Image+Text datasets.**. Missing ratio is 0.5. Image and text modality are evaluated by MSE and PFC respectively. Last two rows are trained with fully-observed data. We show mean and standard deviation over 3 independent runs (lower is better). $\Delta < 0.01$.

We choose CMU-MOSI (Zadeh et al., 2016) and ICT-MMMO (Wöllmer et al., 2013) following Tsai et al. (2019). The author released the features of each modality, and all the numbers are calculated on the feature level. CMU-MOSI (Zadeh et al., 2016) is a collection of 2199 monologue opinion video clips annotated with sentiment. ICT-MMMO (Wöllmer et al., 2013) consists of 340 online social review videos annotated for sentiment. We train all the models using Adam optimizer with learning rate of 1e-3.

|  | Textual-MSE | Acoustic-MSE | Visual-MSE |
|---|---|---|---|
| AE | $0.035 \pm 0.003$ | $0.224 \pm 0.025$ | $0.019 \pm 0.003$ |
| VAE | $0.034 \pm \Delta$ | $0.202 \pm \Delta$ | $0.1273 \pm \Delta$ |
| CVAE w/ mask | $0.43 \pm \Delta$ | $0.257 \pm 0.002$ | $0.020 \pm \Delta$ |
| MVAE | $0.44 \pm \Delta$ | $0.213 \pm 0.001$ | $0.025 \pm \Delta$ |
| VSAE | $\mathbf{0.033 \pm \Delta}$ | $\mathbf{0.200 \pm \Delta}$ | $\mathbf{0.017 \pm \Delta}$ |
| CVAE w/ data | $0.036 \pm \Delta$ | $0.186 \pm \Delta$ | $0.018 \pm \Delta$ |
| VAEAC | $0.042 \pm \Delta$ | $0.257 \pm \Delta$ | $0.019 \pm \Delta$ |

Table 11: **Imputation on CMU-MOSI.** Missing ratio is 0.5. Last two rows are trained with fully-observed data. Evaluated by MSE of each modality. We show mean and standard deviation over 3 independent runs (lower is better). $\Delta < 0.0005$

|  | Acoustic-MSE | Visual-MSE | Textual-MSE |
|---|---|---|---|
| AE | $188.19 \pm 2.083$ | $3.695 \pm 0.004$ | $7.688 \pm 0.243$ |
| VAE | $63.26 \pm 0.757$ | $3.676 \pm 0.103$ | $6.153 \pm 0.232$ |
| CVAE w/ mask | $61.56 \pm 6.584$ | $3.614 \pm 0.015$ | $6.203 \pm 0.423$ |
| MVAE | $174.95 \pm 117.64$ | $\mathbf{3.569 \pm 0.014}$ | $8.501 \pm 3.561$ |
| VSAE | $\mathbf{59.17 \pm 4.120}$ | $\mathbf{3.569 \pm 0.011}$ | $\mathbf{5.108 \pm 0.003}$ |
| CVAE w/ data | $59.22 \pm 11.59$ | $3.367 \pm 0.046$ | $6.398 \pm 0.275$ |
| VAEAC | $78.43 \pm 8.774$ | $3.111 \pm 0.300$ | $18.65 \pm 0.452$ |

Table 12: **Imputation on ICT-MMMO.** Missing ratio is 0.5. Last two rows are trained with fully-observed data. Evaluated by MSE of each modality. We show mean and standard deviation over 3 independent runs (lower is better).

