# OpenReview forum: "Learning from Partially-Observed Multimodal Data with Variational Autoencoders"
_ICLR.cc/2020/Conference — Reject_

### Official Review · AnonReviewer2 · 2019-10-24
**Official Blind Review #2**

**Rating:** 3

**Review:**

Summary:
This paper proposes to impute multimodal data when certain modalities are present. The authors present a variational selective autoencoder model that learns only from partially-observed data. VSAE is capable of learning the joint
distribution of observed and unobserved modalities as well as the imputation mask, resulting in a model that is suitable for various down-stream tasks including data generation and imputation. The authors evaluate on both synthetic high-dimensional and challenging low-dimensional multimodal datasets and show improvement over the state-of-the-art data imputation models.

Strengths:
- This is an interesting paper that is well written and motivated.
- The authors show good results on several multimodal datasets, improving upon several recent works in learning from missing multimodal data.

Weaknesses:
- How multimodal are the datasets provided by UCI? It seems like they consist of different tabular datasets with numerical or categorical variables, but it was not clear what the modalities are (each variable is a modality?) and how correlated the modalities are. If they are not correlated at all and share no joint information I'm not sure how these experiments can represent multimodal data.
- Some of the datasets the authors currently test on are quite toy, especially for the image-based MNIST and SVHN datasets. They should consider larger-scale datasets including image and text-based like VQA/VCR, or video-based like the datasets in (Tsai et al., ICLR 2019).
- In terms of prediction performance, the authors should also compare to [1] and [2] which either predict the other modalities completely during training or use tensor-based methods to learn from noisy or missing time-series data.
- One drawback is that this method requires the mask during training. How can it be adapted for scenarios where the mask is not present? In other words, we only see multiple modalities as input, but we are not sure which are noisy and which are not?

[1] Pham et al. Found in Translation: Learning Robust Joint Representations by Cyclic Translations Between Modalities, AAAI 2019
[2] Liang et al. Learning Representations from Imperfect Time Series Data via Tensor Rank Regularization, ACL 2019

### Post rebuttal ###
Thank you for your detailed answers to my questions.

**Experience Assessment:**

I have published in this field for several years.

**Review Assessment: Checking Correctness Of Derivations And Theory:**

I assessed the sensibility of the derivations and theory.

**Review Assessment: Checking Correctness Of Experiments:**

I carefully checked the experiments.

**Review Assessment: Thoroughness In Paper Reading:**

I read the paper thoroughly.

---

> ### Author Response · Authors · 2019-11-12
> **Reply to Reviewer #2**
>
> (1) Multimodal setting:
> We apologize for not describing experimental settings clearly. In general, we believe multi-modal data is more general than simply image-text or video-text pair. By unifying tabular data also as multi-modal data (with each attribute as one modality), we show that VASE provides us a principled way for imputation and is capable of generalizing to more data families. We update additional multimodal dataset experiments in the point (3) below.
>
> (2) Prediction and Representation learning:
> We consider conducting these experiments during the rebuttal but none of the paper's code has been released by the authors. We agree deep latent variable models explicitly model the data distribution and provide a natural way for representation learning, but in our paper we evaluate the model from the perspective of imputation and generation.
>
> (3) Additional experiments:
> We updated additional imputation experiments on multimodal datasets (see in Appendix C.5) : CMU-MOSI/ICT-MMMO (Tsai et al. 2019), FashionMNIST/MNIST (Wu et al. 2018). Each dataset contains two or three modalities. VSAE outperforms other baselines on multimodal datasets under partially-observed setting.
>
> (4) Require mask during training:
> In our experiments, the binary mask is always fully-observed as is the nature of partially-observed data. A mask simply indicates which  modalities are observed and which are not. We agree that it is very interesting to design a model with partially-observed or even unobserved mask.
> However, it is beyond the scope of this work and we will consider it in future work.
>
>
> [1] Wu et al. Multimodal Generative Models for Scalable Weakly-Supervised Learning, NeurIPS 2018.
> [2] Tsai et al. Learning Factorized Multimodal Representation, ICLR 2019.

---

### Official Review · AnonReviewer4 · 2019-10-30
**Official Blind Review #4**

**Rating:** 3

**Review:**

The paper proposes a novel training method for variational autoencoders that allows using partially-observed data with multiple modalities. A modality can be a whole block of features (e.g., a MNIST image) or just a single scalar feature. The probabilistic model contains a latent vector per modality. The key idea is to use two types of encoder networks: a unimodal encoder for every modality which is used when the modality is observed, and a shared multimodal encoder that is provided all the observed modalities and produces the latent vectors for the unobserved modalities. The whole latent vector is passed through a decoder that predicts the mask of observed modalities, and another decoder that predicts the actual values of all modalities. The “ground truth” values for the unobserved modalities are provided by sampling from the corresponding latent variables from the prior distribution once at some point of training.

While I like the premise of the paper, I feel that it needs more work. My main concern is that sampling the target values for the unobserved modalities from the prior would almost necessarily lead to blurry synthetic “ground truth” for these modalities, which in turn means that the model would produce underconfident predictions for them. The samples from MNIST in Figure 3 are indeed very blurry, supporting this. Furthermore, the claims of the model working for non-MCAR missingness are not substantiated by the experiments. I believe that the paper should currently be rejected, but I encourage the authors to revise the paper.

Pros:
* Generative modelling of partially observed data is a very important topic that would benefit from fresh ideas and new approaches
* I really like the idea of explicitly modelling the mask/missingness vector. I agree with the authors that this should help a lot with non completely random missingness.

Cons:
* The text is quite hard to read. There are many typos (see below). The text is over the 8 page limit, but I don’t think this is justified. For example, the paragraph around Eqn. (11) just says that the decoder takes in a concatenated latent vector. The MNIST+SVHN dataset setup is described in detail, yet there is no summary of the experimental results, which are presented in the appendix.
* The approach taken to train on partially-observed data is described in three sentences after the Eqn. (10). The non-observed dimensions are imputed by reconstructions from the prior from a partially trained model. I think that this is the crux of the paper that should be significantly expanded and experimentally validated. It is possible that due to this design choice the method would not produce sharper reconstructions than the original samples from the prior. Figures 3, 5 and 6 indeed show very blurry samples from the model. Furthermore, it is not obvious to me why these prior samples would be sensible at all, given that all modalities have independent latents by construction.
* The paper states multiple times that VAEAC [Ivanov et al., 2019] cannot handle partially missing data, but I don’t think this is true, since their missing features imputation experiment uses the setup of 50% truly missing features. The trick they use is adding “synthetic” missing features in addition to the real ones and only train on those. See Section 4.3.3 of that paper for more details.
* The paper states that “it can model the joint distribution of the data and the mask together and avoid limiting assumptions such as MCAR”. However, all experiments only show results in the MCAR setting, so the claim is not experimentally validated.
* The baselines in the experiments could be improved. First of all, the setup for the AE and VAE is not specified. Secondly, it would be good to include a GAN-based baseline such as GAIN, as well as some more classic feature imputation method, e.g. MICE or MissForest.
* The experiments do not demonstrate that the model learns a meaningful *conditional* distribution for the missing modalities, since the provided figures show just one sample per conditioning image.

Questions to the authors:
1. Could you comment on the differences in your setup in Section 4.1 compared to the VAEAC paper? I’ve noticed that the results you report for this method significantly differ from the original paper, e.g. for VAEAC on Phishing dataset you report PFC of 0.24, whereas the original paper reports 0.394; for Mushroom it’s 0.403 vs. 0.244. I’ve compared the experimental details yet couldn’t find any differences, for example the missing rate is 0.5 in both papers.
2. How do you explain that all methods have NRMSE > 1 on the Glass dataset (Table 1), meaning that they all most likely perform worse than a constant baseline?

Typos and minor comments:
* Contributions (1) and (2) should be merged together.
* Page 2: to literature -> to the literature
* Page 2: “This algorithm needs complete data during training cannot learn from partially-observed data only.”
* Equations (1, 2): z and \phi are not consistently boldfaced
* Equations (4, 5): you can save some space by only specifying the factorization (left column) and merging the two equations on one row
* Page 4, bottom: use Bernoulli distribution -> use factorized/independent Bernoulli distribution
* Page 5, bottom: the word “simply” is used twice
* Page 9: learn to useful -> learn useful
* Page 9: term is included -> term included
* Page 9: variable follows Bernoulli -> variable following Bernoulli
* Page 9: conditions on -> conditioning on

**Experience Assessment:**

I have published one or two papers in this area.

**Review Assessment: Checking Correctness Of Derivations And Theory:**

I assessed the sensibility of the derivations and theory.

**Review Assessment: Checking Correctness Of Experiments:**

I assessed the sensibility of the experiments.

**Review Assessment: Thoroughness In Paper Reading:**

I read the paper at least twice and used my best judgement in assessing the paper.

---

> ### Author Response · Authors · 2019-11-12
> **Reply to Reviewer #4**
>
> (1) Reconstruction from prior during training:
> The crux of the proposed model is the selective proposal distribution. "Pseudo" sampling for unobserved modalities during training provides a way to facilitate model training process. We evaluated the model under two training settings: (I) optimize the final ELBO without conditional log-likelihood for unobserved modalities x_u; and (II) optimize the final ELBO with  conditional log-likelihood of unobserved modalities. This is realized by utilizing the "pseudo" sampling described before (and in the paper).
> The results are comparable but the added term in setting II shows benefits on some datasets. While setting I is solely based on the observed modalities, the setting II incorporates the unobserved modalities along with the observed ones. By using the complete data, the setting II describes the complete ELBO corresponding to the partially observed multimodal data (in consideration).
>
> (2) Comparison with VAEAC:
> In order to establish fair comparison, we used the same backbone network structures and training criteria for all baseline models and our proposed VSAE. Therefore, the implementation details differ from the original VAEAC paper. We did our best to maintain the optimization details described in all baseline papers.
> Experiments on VAEAC with partially-observed data are also conducted. Results show that VAEAC under this setting can achieve comparable performance on categorical datasets: 0.245(0.002) on Phishing, 0.399(0.011) on Mushroom while the errors of VSAE are 0.237(0.001) on Phishing,  0.396(0.008) on Mushroom. However, on numerical and bimodal datasets, partially trained VAEAC performs worse than VSAE :
> *VSAE:
> 0.455(0.003) on Yeast; 1.312(0.021) on Glass;0.1376(0.0002) on MNIST+MNIST; 0.1198(0.0001) on MNIST+SVHN;
> *VAEAC trained partially:
> 0.878(0.006) on Yeast; 1.846(0.037) on Glass;0.1402(0.0001) on MNIST+MNIST; 0.2126(0.0031) on MNIST+SVHN.
>
> (3) Experiments under synthetic non-MCAR masking:
> As mentioned by the reviewer, we conduct experiments on non-MCAR masking following state-of-the-art non-MCAR model MIWAE [2]. Same as MIWAE, we synthesize masks by defining some rules to specify the probability of a Bernoulli distribution. Please refer to Table 3 and Appendix C.4 for updated comparison results. VSAE outperforms MIWAE under all MCAR, MAR and NMAR masking mechanisms.
>
> (4) Baselines:
> All baselines considered in the paper are designed to have comparable number of parameters (same or larger than our model) to make the comparison fair. We have updated the baseline details in the Appendix B.3. Although GAN-based models show promising imputation results, they usually fail to model data distribution properly. Therefore, we do not consider them as our baseline models. It is also important to note that VSAE is not a model designed only for imputation, but a generic framework to learn from partially-observed data for both imputation and generation.
>
> (5) Conditional imputation:
> When performing imputation, we assume that the generation is not conditioned on the observed image, but only conditioned on the factorized latent variables. Input an observed image to the model, we observe a "conditional" distribution if we independently sample from the latent variables. See Figure.7 in updated Appendix C.2.
>
> (6) Answers to the questions:
> 1. Please refer to point (2) for detailed explanation on comparison with VAEAC. In summary, there are multiple reasons why the performance is not identical with the original VAEAC: (I) the back-bone structures are not the same; (II) training criteria (including batch size, learning rate, etc.) are not the same; and (III)  training/validation/test split is different. We would like to emphasize that the aforementioned changes are necessary to establish fair comparison.
>
> 2.  We adopt the calculation from [1] where NRMSE is RMSE normalized by the standard deviation of each feature followed by an average over all imputed features. The standard deviation of ground truth features does not guarantee NRMSE < 1.
>
>
> [1] Ivanov et al.Variational Autoencoder with Arbitrary Conditioning, ICLR 2019
> [2] Mattei et al. MIWAE: Deep Generative Modelling and Imputation of Incomplete Data Sets, ICML 2019

---

> > ### Comment · AnonReviewer4 · 2019-11-14
> > **Thank you for a detailed reply!**
> >
> > I would like to thank the authors for their detailed reply and the additional experiments they ran. Given a good rebuttal and improvements in the submission, I am increasing the score to weak reject, yet I still think that the manuscript is not well suited for publication in the current form. However, I encourage the authors to improve the paper and resubmit.
> >
> > Here are my comments on the reply:
> >
> > (1) Reconstruction from prior during training
> > > (I) optimize the final ELBO without conditional log-likelihood for unobserved modalities x_u
> > In this alternative setting you do not get any training signal on how to reconstruct the missing modalities in the decoder. This is why I say that the way you provide the ground truth is the crucial ingredient of the model.
> > There is no response on the blurriness of the samples.
> >
> > (2) Comparison with VAEAC
> > - The latest revision of the text still states that VAEAC cannot work with the partially-observed data: “VAE with arbitrary conditioning (VAEAC) which allows generation of missing data conditioned on any combination of observed data. This algorithm needs complete data during training and cannot learn from partially-observed data only.”
> > -Given your results, it seems that VAEAC is competitive in the partially-observed setting, and so should probably be added as a baseline.
> > - I still don’t understand why the VAEAC results are better on some sets and worse on others. More generally, changing the experimental setting compared to previous works without a clear reason is not a good scientific practice.
> >
> > (3) Experiments under synthetic non-MCAR masking
> > (5) Conditional imputation:
> > Thank you for adding these experiments. I believe they strengthen the paper by showing that the model can handle non-MCAR masking and produce a diverse set of samples given the observed modalities.
> >
> > (4) Baselines:
> > I don’t find the response on GANs convincing. The paper uses metrics such as RMSE which do not really require the model to produce diverse samples, so GANs seem like a good baseline. Furthermore, there is no reply on the non-deep learning baselines.
> >
> > (6.2) NRMSE > 1
> > I did not understand this argument. Surely NRMSE can be arbitrarily large, but a well-tuned algorithm should probably obtain NRMSE < 1, i.e. perform better than a constant predictor set to the true mean. VAEAC paper reports NRMSE of 0.87-0.91 for Glass, which is consistent with this.

---

### Official Review · AnonReviewer1 · 2019-11-01
**Official Blind Review #1**

**Rating:** 3

**Review:**

The paper proposed variational selective autoencoders (VSAE) to learn from partially-observed multimodal data.  Overall, the proposed method is elegant; however, the presentation, the claim, and the experiments suffer from significant flaws. See below for detailed comments.

[Pros]
1. The main idea of the paper is to propose a generative model that can handle partially-observed multimodal data during training. Specifically, prior work considered non-missing data during training, while we can't always guarantee that all the modalities are available. Especially in the field of multimodal learning, we often face the issue of imperfect sensors. This line of work should be encouraged.

2. In my opinion, the idea is elegant. The way the author handles the missingness is by introducing an auxiliary binary random variable (the mask) for it. Nevertheless, its presentation and Figure 1 makes this elegant idea seems over-complicated.
[Cons]

1. [The claim] One of my concerns for this paper is the assumption of the factorized latent variables from multimodal data. Specifically, the author mentioned Tsai et al. assumed factorized latent variables from the multimodal data, while Tsai et al. actually assumed the generation of multimodal data consists of disentangled modality-specific and multimodal factors. It seems to me; the author assumed data from one modality is generated by all the latent factors (see Eq. (11)), then what is the point for assuming the prior of the latent factor is factorized (see Eq. (4) and (5))? One possible explanation is because we want to handle the partially-observable issues from multimodal data, and it would be easier to make the latent factors factorized (see Eq. (6)). The author should comment on this.

2. [Phrasing.] There are too many unconcise or informal phrases in the paper. For example, I don't understand what does it mean in "However, if training data is complete, ..... handle during missing data during test." Another example would be the last few paragraphs on page 4; they are very unclear. Also, the author should avoid using the word "simply" too often (see the last few paragraphs on page 5).

3. [Presentation.] The presentation is undesirable. It may make the readers hard to follow the paper. I list some instances here.
		a. In Eq. (3), it surprises me to see the symbol \epsilon without any explanation.
		b. In Eq. (6), it also surprises me to see no description of \phi and \psi. The author should also add more explanation here, since Eq. (6)  stands a crucial role in the author's method.
		c. Figure 1 is over-complicated.
		d. What is the metric in Table 1 and 2?  The author never explains. E.g., link to NRMSE and PFC to the Table.
		e. What are the two modalities in Table 2? The author should explain.
		f. The author completely moved the results of MNIST-SVHN to Supplementary. It is fine, but it seems weird that the author still mentioned the setup of MNIST+SVHN in the main text.
		g. The author mentioned, in Table , the last two rows serve the upper bound for other methods. While some results are even better than the last two rows. The author should explain this.
		h. Generally speaking, the paper does require a significant effort to polish Section 3 and 4.

4. [Experiments.] The author presented a multimodal representation learning framework for partially-observable multimodal data, while the experiments cannot corraborrate the claim. First, I consider the tabular features as multi-feature data and less to be the multimodal data. Second, the synthetic image pairs are not multimodal in nature. These synthetic setting can be used for sanity check, but cannot be the main part of the experiments. The author can perhaps consider the datasets used by Tsai et al. There are seven datasets, and they can all be modified to the setting of partially-observable multimodal data. Also, since the synthetic image pairs are not multimodal in nature, it is unclear to me for what the messages are conveyed in Figure 3 and 4.


I do expect the paper be a strong submission after a significant effort in presentation and experimental designs. Therefore, I vote for weak rejection at this moment.

**Experience Assessment:**

I have published in this field for several years.

**Review Assessment: Checking Correctness Of Derivations And Theory:**

I carefully checked the derivations and theory.

**Review Assessment: Checking Correctness Of Experiments:**

I carefully checked the experiments.

**Review Assessment: Thoroughness In Paper Reading:**

I read the paper thoroughly.

---

> ### Author Response · Authors · 2019-11-12
> **Reply to Reviewer #1**
>
> We would like to thank the reviewer for providing valuable and detailed feedback. We have addressed the clarity concerns in the updated paper. Figure captions, metrics used in the table, etc, as mentioned in the presentation section of the review have been carefully examined and updated in the paper.
> We will reorganize the experiment section to better present the comparisons under different experimental settings.
>
> (1) Factorized Latent Variables:
> The factorization of latent space with respect to the modalities provides a way to differentiate observed and unobserved modalities. Therefore, VSAE is capable of handling partially-observed data where the missing modalities can be arbitrary. In addition, the embeddings are intuitively more meaningful as input to unimodal encoders is now limited to only observed modalities, eliminating the effect of missing modalities.
> When performing imputation/generation, however, we want to capture the dependencies between modalities. In other words, unobserved modalities should be imputed based on the information extracted from observed modalities. For experiments, we design this by conditioning decoders on all latent variables, essentially accessing information from all observed modalities. This is not in contradiction to the factorized latent variable assumption. Instead, the encoders try to embed each modalities individually, while decoders learn the dependencies between different modalities.
>
> (2) Multimodal Experiments:
> We apologize for unclear description of experimental settings. In general, we believe multi-modal data is more general than conventional image-text or video-text pairs. By unifying tabular data also as multi-modal (with each attribute as one modality), we show that VSAE provides us a principled way for imputation, capable of generalizing to more data families. Specifically, we conducted experiments on two types of data:
> (1) low-dimensional tabular data, and (2) high-dimensional data (pixel or text) as "multimodal" to better define the overall task of learning from partially-observed data.
> Upon request, we have included more extensive experiments following [1] on MNIST/FashionMNIST, and [2] on CMU-MOSI/ICT-MMMO. Results are reported in Table 10 and Table 11 (Appendix C.5). As shown, VSAE consistently outperforms baseline models across the added experiments as well.
>
> (3) Discussions on Comparison with Upper Bound Methods:
> Models trained with fully-observed data in theory should have better performance, thus we treat them as upper bound methods. However, it is very interesting to observe that in some cases, VSAE have superior performances. One possible explanation is that missing modalities introduces extra noise into the model as regularizer, thereby, increasing the generalization ability. However, detailed experiments and more discussions need to be carried out to back up this explanation.
>
>
> [1] Wu et al. Multimodal Generative Models for Scalable Weakly-Supervised Learning, NeurIPS 2018.
> [2] Tsai et al. Learning Factorized Multimodal Representation, ICLR 2019.

---

> > ### Comment · AnonReviewer1 · 2019-11-14
> > **Response to the Reply**
> >
> > I like the method and the problem to tackle. I went through the response, and I appreciated that the author had tried to address the comments.
> >
> > However, the two main problems are remaining:
> > 1) I don't feel comfortable if you called low-dimensional tabular data as multimodal data. I think the presentation before the experimental section is over-claimed, but the experiments should be more comprehensive. Besides, why separating the three modalities in CMU-MOSI/ ICT-MMMP? It feels weird to me since the author claimed the method should be multimodal.
> >
> > 2) The second one is more serious. I don't feel the presentation flow of the paper meets the bar for a top conference paper. I do think after a significant effort in the presentation, the quality of the paper can be significantly improved.

---

### Official Review · AnonReviewer5 · 2019-11-03
**Official Blind Review #5**

**Rating:** 6

**Review:**

This paper proposes variational selective autoencoders (VSAE) to learn the joint distribution model of full data (both observed and unobserved modalities) and the mask information from arbitrary partial-observation data. To infer latent variables from partial-observation data, they introduce the selective proposal distribution that switches encoders depending on whether each input modality is observed.

This paper is well written, and the method proposed in this paper is nice. In particular, the idea of the selective proposal distribution is interesting and provides an effective solution to deal with the problem of missing modality in conventional multimodal learning. The experiment is also well structured and shows higher performance than the existing models.  However, I have some questions and comments, so I’d like you to answer them.

Comments:
- The authors state that x_j is sampled from the "prior network" to calculate E_x_j in Equation 10, but I didn’t understand how this network is set up. Could you explain it in detail?
- The authors claim that adding p(m|z) to the objective function (i.e., generating m from the decoder) allows the latent variable to have mask information. However, I don’t know how effective this is in practice. Specifically, how performance differs compared to when p (m | z) is not used and the decoder p (x | z, m) is conditioned by the mask included in the training set instead of the generated mask?
- Why did you not do image inpainting in higher-dimensional experiments like Ivanov et al. (2019), i.e., considering each pixel as a different modality? Of course, I know that Ivanov et al. require the full data as input during training, but I’m interested in whether VSAE can perform inpainting properly even if trained given imperfect images.

**Experience Assessment:**

I have published one or two papers in this area.

**Review Assessment: Checking Correctness Of Derivations And Theory:**

I carefully checked the derivations and theory.

**Review Assessment: Checking Correctness Of Experiments:**

I carefully checked the experiments.

**Review Assessment: Thoroughness In Paper Reading:**

I read the paper thoroughly.

---

> ### Author Response · Authors · 2019-11-12
> **Reply to Reviewer #5**
>
> (1) Prior Network:
> During training phase, we sample from prior network to generate "pseudo" observations for unobserved modalities. The pseudo observations are then used to estimate the conditional likelihood for such modalities (E_x_j in the ELBO).
> Practically, we follow a two-stage method in our implementation. At each iteration, the first stage imputes unobserved modalities (with latent code sampled from approximate posterior for observed modalities, and prior for unobserved modalities), followed by the second stage to estimate ELBO based on the imputation and backpropagate corresponding gradients.
>
> (2) Conditioning on Ground-Truth Mask:
> We conduct experiments with decoder p(x|z, m) conditioned on the original mask in training set, and observe comparable performance and convergence time. The mask distribution might be easier to learn as compared to data distribution (since the mask is fully-observed). However, we argue that jointly learning the mask distribution and data distribution provides us an opportunity to further analyze the missing mechanism and potentially can facilitate other down-stream tasks.
>
> (3) Image Inpainting:
> We appreciate the reviewer's suggestion on evaluate the effectiveness of our model on image inpainting task.
> However, with our current setup, an encoder is trained for each modality respectively, making it difficult to scale to inpainting task, if we treat each pixel as an individual modality.
> Nevertheless, we believe this is an interesting extension. The backbone models and mathematical formulations can be very similar, if not the same. A potential solution could be to employ patch level encoders to reduce the total number of encoders needed.

---

### Author Response · Authors · 2019-11-12
**Reply to Reviewers**

We would like to thank all reviewers for their thorough and valuable feedback. We discuss the questions and concerns below and provide clarifications in the updated paper.

---

### Decision · Program_Chairs · 2019-12-19

**Decision:**

Reject

**Comment:**

This submission proposes a VAE-based method for jointly inferring latent variables and data generation. The method learns from partially-observed multimodal data.

Strengths:
-Learning to generate from partially-observed data is an important and challenging problem.
-The proposed idea is novel and promising.

Weaknesses:
-Some experimental protocols are not fully explained.
-The experiments are not sufficiently comprehensive (comparisons to key baselines are missing).
-More analysis of some surprising results is needed.
-The presentation has much to improve.

The method is promising but the mentioned weaknesses were not sufficiently addressed during discussion. AC agrees with the majority recommendation to reject.